# Preparation, Structure, and Properties of PVA–AgNPs Nanocomposites

**DOI:** 10.3390/polym15020379

**Published:** 2023-01-10

**Authors:** Oksana Velgosova, Lívia Mačák, Erika Múdra, Marek Vojtko, Maksym Lisnichuk

**Affiliations:** 1Institute of Materials and Quality Engineering, Faculty of Materials Metallurgy and Recycling, Technical University of Kosice, Letná 9/A, 042 00 Kosice, Slovakia; 2Division of Ceramic and Non-Metallic Systems, Institute of Materials Research, Slovak Academy of Sciences, Watsonova 47, 040 01 Kosice, Slovakia; 3Faculty of Science, Institute of Physics, Pavol Jozef Šafárik University in Košice, Park Angelinum 9, 040 01 Košice, Slovakia

**Keywords:** nanoparticles, silver, polymer matrix nanocomposite, PVA, “ex situ” preparing

## Abstract

The aim of the work was to prepare a polymer matrix composite doped by silver nanoparticles and analyze the influence of silver nanoparticles (AgNPs) on polymers’ optical and toxic properties. Two different colloids of AgNPs were prepared by chemical reduction. The first colloid, a blue one, contains stable triangular nanoparticles (the mean size of the nanoparticles was ~75 nm). UV–vis spectrophotometry showed that the second colloid, a yellow colloid, was very unstable. Originally formed spherical particles (~11 nm in diameter) after 25 days changed into a mix of differently shaped nanoparticles (irregular, triangular, rod-like, spherical, decahedrons, etc.), and the dichroic effect was observed. Pre-prepared AgNPs were added into the PVA (poly(vinyl alcohol)) polymer matrix and PVA–AgNPs composites (poly(vinyl alcohol) doped by Ag nanoparticles) were prepared. PVA–AgNPs thin layers (by a spin-coating technique) and fibers (by electrospinning and dip-coating techniques) were prepared. TEM and SEM techniques were used to analyze the prepared composites. It was found that the addition of AgNPs caused a change in the optical and antibiofilm properties of the non-toxic and colorless polymer. The PVA–AgNPs composites not only showed a change in color but a dichroic effect was also observed on the thin layer, and a good antibiofilm effect was also observed.

## 1. Introduction

Currently, nanotechnology is one of the fastest moving sectors; in particular, the synthesis and application of metal nanoparticles are at the forefront of interest. Metal nanoparticles are used in various industries, and due to this, the demand for nanoparticles continues to rise. There are several methods to prepare nanoparticles; conventional methods include physical and chemical methods [1]. Recently, many scientists have been working on green (biological) methods that use microorganisms or plants and their parts to reduce metal particles [2,3]. New, ecologically acceptable methods include mechanochemical methods, which use milling and chemical reduction for nanoparticle synthesis [4], and photochemical methods [5]. Of all the available methods, the chemical method can be considered the most suitable, when we ignore the possible ecological problems. By chemical reduction, it is possible to prepare colloidal nanoparticles of any metal, and what is most important, the chemical synthesis route enables us to take control of the shape and size of nanoparticles.

Noble metal nanoparticles are well known because of their exceptional properties that result from their nano-size. Noble metal nanoparticles are strong absorbers and scatterers of visible light due to localized surface plasmon resonance (LSPR). Because of these singular optical features, noble metal nanoparticles stimulate great interest in implementation in photonics. For instance, nanoclusters composed of 2–8 silver atoms could be the basis for a new type of optical data storage. Moreover, fluorescent emissions from the clusters could potentially be used in biological labels and electroluminescent displays [6,7]. Under certain conditions, silver nanoparticle colloids display a dichroic effect. The dichroic effect has wide applications in art and optical devices such as lenses, beam steering, and dichroic filters (designed to selectively allow wavelengths within a specific transmission band to pass through while reflecting all other colors). Not only the optical but also the electrochemical properties of AgNPs are studied because of their application in sensors [8,9].

Silver is also known for its good toxic properties, which are accentuated by the nanoscale. Silver is therefore widely used in the field of disinfection, e.g., in healthcare, or in sanitation products. In the case of, e.g., gold, the impact of size decrease is more significant; non-toxic Au as a bulk material becomes toxic when its size is reduced to the nano level [10,11].

The chemical reduction process, usually, synthesizes nanoparticles in polar on nonpolar solutions, and the result is a colloid solution of nanoparticles. In this form, nanoparticles have limited use, but if they are incorporated into the matrix, the possibilities of their applications will increase significantly. Polymers, which are often used in medicine and the packaging industry, appear to be the most suitable matrices. Aseptic coatings and thin layers prepared based on polymers containing Ag nanoparticles are also very promising. There are many ways to incorporate nanoparticles into the polymer matrix. Generally, they can be divided into two main groups: in situ [12,13] and ex situ [14,15].

The difference between these methods is in the way the nanoparticles were added into the polymer matrix; in the in situ methods, nanoparticles are formed directly in the polymer matrix, from the entry substances that react together. In the case of the production of polymer composite by the ex situ method, pre-prepared nanoparticles with a known size and shape and of a required quantity are added to the prepared matrix. The development in nanotechnology, membrane, and thin film sciences requires the use of a variety of polymer matrixes. It can be seen, that polymers such as poly(vinyl alcohol) (PVA) and poly(N-vinyl 2-pyrrolidone) (PVP) [16], polysulfone (PS) [17], polyethersulfone (PES) [18], polyvinylidene fluoride (PVDF) [19], cellulose acetate (CA) [20], etc., are often used. The choice of the polymer depends on the final use since the physicochemical properties of the polymer influence the structure of composites obtained at the end of the phase inversion process [21]. The addition of a secondary phase (silver nanoparticles) into the polymer matrix changes the composite’s final properties. In many studies, the effect of the AgNPs on the final properties has been studied. The incorporation of AgNPs into the PES matrix significantly improved the antibacterial properties [22] and thermal stability of the polymer composite [23]. AgNPs also have the ability to increase the hydrophilic properties of a composite [24]. In general, a non-toxic polymer can obtain toxic properties, and metal nanoparticles can transfer optical (e.g., filters of different colors can be prepared) and conductance properties into the polymer. For polymer composite final properties, the most crucial parameter is the distribution of nanoparticles in the matrix. In addition, the matrix morphology is important; it is clear, that porous matrices and matrices with the ability to gradually release nanoparticles are desired for membrane and thin film applications in disinfection fields [25,26]. A lot of work was carried out in this field but in spite of that, a lot of experiments have to be carried out to improve the final method of composite production with respect to their properties.

The aims of this work were to prepare silver nanoparticles of different shapes and incorporate them into the polymer matrix (PVA) to observe the synergic effect of the polymer and AgNPs. Synergistic effects are non-linear cumulative effects of two components that are combined in one material, for example in a composite. In the case of PVA (nontoxic and colorless) and AgNPs’ (which have interesting optical properties and which are also well known for their toxicity) connection, unique properties can be achieved. The ex situ method of AgNPs’ incorporation was used. Subsequently, the polymer composite thin films were produced by the spin-coating technique, and the fibers (non-woven textiles) were produced by electrospinning and dip-coating. The influence of different techniques on the distribution of nanoparticles was observed. The ability of AgNPs to transform their antibacterial and optical properties into PVA–AgNPs composite was proven. TEM and SEM techniques and UV–vis spectrophotometry were used as the main techniques for nanoparticles and PVA–AgNPs composite analysis.

## 2. Materials and Methods

### 2.1. Materials

As a silver precursor, silver nitrate (>98%) purchased from Mikrochem Ltd., Pezinok, Slovakia, was used. Sodium borohydride (≥98%), sodium citrate (TSC) (≥ 99%), hydrogen peroxide (30%), polyvinylpyrrolidone (PVP) M.W. approx. 360,000, and poly(vinyl alcohol) (PVA) M.W. approx. 146,000 were also purchased from Mikrochem Ltd., Pezinok, Slovakia, and used as received. De-ionized water was used for preparing all of the solutions.

### 2.2. Synthesis of Silver Nanoparticles

Silver nanoparticles were synthesized by chemical methods by reduction of Ag^+^ ions to Ag^0^. The solutions labeled B and I were prepared; the amount and concentrations of the reagents used for both solutions are in Table 1.

The preparation procedure of solution B: at first a solution of silver nitrate (4300 mL, 0.11 mM solution) was prepared. Subsequently, 368 mL of 30 mM sodium citrate solution, 368 mL of 2% *w*/*w* polyvinylpyrrolidone, 12 mL 30% *w*/*w* of hydrogen peroxide, and 20 mL of 0.1 M sodium borohydride were added at constant steering. Solution I was prepared in the same way using the reagents according to Table 1.

The prepared colloidal solutions were centrifuged (14,000 rpm for 30 min) to concentrate the nanoparticles. The final (after centrifugation) volume of solutions was 36 mL. Concentrate nanoparticles were used for PVA–AgNPs composite preparations.

### 2.3. “Ex Situ” Preparation of Polymer Matrix Composites

Polymer matrix composites doped by silver nanoparticles were produced by the exsitu method. Polyvinyl alcohol matrix (4 g) was mixed with a prepared colloid solution of AgNPs (36 mL) and stirred at 450 rpm at 85 °C for 2 h (8% solution of PVA was prepared). After stirring PVA–AgNPs composite was ultrasonicated for 15 min. Subsequently, a composite solution of PVA–AgNPs was used to prepare the thin layers and nanofibers.

The thin layers were prepared by the spin-coating technique; the prepared, thin layers were allowed to dry at ambient temperature. The PVA–AgNPs composite nanofibers were prepared by two methods:The prepared PVA–AgNPs composite solution was treated by a needleless electrospinning technique (Nanospider), and non-woven fabrics of PVA fibers doped by AgNPs were prepared.The pure PVA solution (8% solution: PVA powder + H_2_O) was electrospun, and nanofibers in the form of non-woven fabric were prepared. Subsequently, non-woven fabric was dip-coated by AgNPs. The procedure was as follows: the non-woven fabric of PVA was immersed into a prepared AgNPs colloid solution and dried in a dryer at 45 °C for 30 min; immersion and drying were repeated five times.

The distribution of nanoparticles in the PVA matrix and the influence of different techniques on AgNPs’ incorporation on their location in the matrix were analyzed.

### 2.4. Methods

Synthesized AgNPs were monitored by a UV–vis spectrometer (UNICAM UV-vis Spectrophotometer UV4). The size and morphology of the nanoparticles were studied by means of TEM (JEOL model JEM-2000FX, an accelerating voltage of 200 kV). Image analysis (ImageJ software) was used for the analysis of Ag nanoparticles’ size distribution. The morphology of AgNPs was observed by scanning electron microscopy SEM/FIB (SEM/FIB ZEISS-AURIGA Compact).

The PVA–AgNPs composite fibers and fibers of pure PVA were prepared by needle-less electrospinning technology (Nanospider). The applied voltage was 82 kV and the distance between spinning and collector electrodes was 150 mm. The setting of the electrospinning condition influences the fibers’ diameter. The PVA–AgNPs composite layers were prepared by spin-coating techniques, by a Spin Coater SCC-200. The presence and distribution of nanoparticles in the polymer composites (thin films and nanofibers) were analyzed by scanning electron microscopy (SEM).

The antimicrobial activity of colloidal AgNPs, PVA–AgNPs composite layers, and fibers was evaluated using the standard disk-diffusion method with some modifications. Agar plates in Petri dishes were inoculated with algal. The 15 µL of prepared colloidal AgNPs were plated on agar plates using sterile swabs. Samples of layers and fibers of the PVA–AgNPs composite were cut from the prepared samples and, after sterilization under a germicidal lamp, they were applied to agar plates. The agar plates were incubated at room temperature. As a control, a sample of pure PVA fibers was used. The presence and size of the inhibition zone on the agar plates were checked after 14 days of growth.

## 3. Results and Discussion

Chemical synthesis of silver nanoparticles has, compared to physical and biological methods, several advantages, one of which is the rate of reactions. After adding all of the reactants, the change of color starts immediately. Ordinarily, transparent solutions change to blue in the case of solution B and yellow for solution I, Figure 1.

Thanks to local surface plasmon resonance (LSPR), which in the case of metal nanoparticles is the resonant oscillation of conduction electrons in response to incident light [3], it is possible to observe not only different colors of solutions, but UV–vis spectrophotometry can confirm the presence of nanoparticles in the solution and show the shape and size, Figure 2a.

Based on the results of UV–vis spectroscopy, it can be concluded that solution B was very stable, Figure 2a. No changes were observed in the shape and position of the LSPR band and also in the position of ABS_max_ for more than three weeks, Figure 2a. In addition, the color of the solution did not change over time. The presence of a distinctive peak around 850 nm, a flat peak (more like a shoulder) around 500 nm, and a sharp peak at 345 nm indicates the presence of triangular nanoparticles. This assumption was confirmed by TEM and SEM analysis, Figure 2b,c. Based on TEM micrographs, the average size of the nanoparticles was ~75 nm. SEM micrographs confirmed the medium size and triangular shape of the nanoparticles and based on the SEM, the thickness of the triangular prisms, of which less than 3 nm could be measured.

On the other hand, the LSPR band (solution I) recorded significant changes not only in the shape but also in the position of the ABS_max_ peaks and their number, Figure 3a. There was one peak at D0 (ABS_max_ at wavelength λ_max_ = 392 nm). A single peak at a wavelength of around 400 nm indicates the presence of uniform and spherical nanoparticles. This assumption was confirmed by TEM analysis, Figure 3b. The mean particle diameter at D0 was ~11 nm.

Over the course of 25 days, the LSPR band of sample I changed dramatically, as shown in Figure 4. The position of the ABS_max_ peak at ~400 nm (D0) did not significantly change on D5 (λ_max_ = 402 nm), but a rise of the shoulder at λ = 630 nm was evident. This shoulder continues to rise and on D10 there was also an evident strong second peak at 630 nm, a decrease of the peak at 400 nm, and a small peak at 310 nm. Such changes in LSPR band shape indicate the changes in nanoparticles’ shape and size. Based on our experiences and the results of other authors it can be concluded that the presence of more than one peak, indicates the presence of a mix of nanoparticles (irregular, triangular, rod-like, spherical, decahedrons, etc.). It is obvious that solution I was not stable even after D20, where four peaks can be observed, Figure 4. The LSPR changes were also reflected in the color of the colloids, Figure 4, which changed from yellow through light green, and turquoise, to different shades of blue. Finally, on D25, the color changed to violet, and the dichroic effect occurred, Figure 4. TEM analysis on D25, Figure 3c,d, confirmed the presence of differently shaped nano-particles (triangular nano-prisms, rod-like nanoparticles, spherical, etc.).

Dichroic solutions show different colors when viewed from different directions. The reason for such a phenomenon is the different absorption coefficients for light polarized in different directions. In our case, the reflected light view is green, while the transmitted light solution appears violet, Figure 4. The dichroic effect is very rare; achieving and maintaining such special conditions in a solution requires very precise preparation. There are only a few works describing the dichroic effect of Ag and Au nanoparticles [27,28]. Dichroism is the result of the presence of the different shapes and sizes of nanoparticles in the colloids. A very important factor here is also the proportional representation of individual shapes.

It is clear that H_2_O_2_ is a key factor in modifying the shape of nanoparticles. H_2_O_2_ as an oxidant has no reducing ability, but the combination of TSC+PVP+ H_2_O_2_ +NaBH_4_ ensures the formation of triangular nanoparticles, and such nanoparticles remain stable for a long time. Without H_2_O_2_, spherical nanoparticles with poor stability are formed, leading to changes in shape and size.

To confirm the presence of silver, energy-dispersive X -ray spectroscopy was used. Analyses were performed on both samples, Figure 5. It is evident that a strong silver signal in both samples B and I is presented.

### 3.1. Polymer Matrix Composite

Synthesized triangular nanoparticles (sample B) and dichroic nanoparticles (sample I) were used to prepare the polymer composites. Silver nanoparticles are well known for their antibacterial and antibiofilm properties [29,30]. The optical properties of AgNPs are also very interesting. We assume that not only the antibacterial but also the optical properties (different colors and dichroism) of synthesized nanoparticles can be transferred into the polymer matrix. PVA is a widely used, non-toxic, odorless, and colorless polymer. It is used especially in medicine (catheters, cartilage, and contact lenses), but it can also be found in the production of different layers and 3D printing. The polymer composite of PVA–AgNPs could show antibacterial/antibiofilm properties, and the use of such composites could significantly widen, especially in the water management industry or in some applications in medicine. If PVA gains the color properties of AgNPs, colored polymers can find application in optics filters for the selective passing of light. Dichroic filters are popular in architectural and theatrical applications.

Thin layers of PVA–AgNPs by spin-coating and fibers of PVA–AgNPs by electrospinning and dip-coating techniques were prepared. Figure 6a,b shows blue and dichroic thin layers prepared from samples B and I, respectively.

Thin layers, which contain triangular nanoparticles become blue, Figure 6a, and those containing dichroic nanoparticles, become green in reflected light and violet/beige in transmitted light, Figure 6b. Figure 6c,d shows fibers prepared from sample B by electrospinning and dip-coating of pure PVA fibers. It is clear that the triangular nanoparticles colored the fibers blue, but the color of the fibers is not as bright as it was in the case of a thin layer. In the case of dichroic nanoparticles, the color of the fibers was in the shade of violet/beige (data not shown) and the dichroic effect did not occur.

Figure 7 shows the details of the nanoparticle distribution in the PVA–AgNPs’ thin layer (sample B). It is clear that AgNPs formed clusters in the PVA matrix, which consists of a large number of triangular nanoparticles. The same behavior (cluster formation) of the nanoparticles was observed in sample I, (data not shown). It is obvious that the 8% PVA solution was too dense, and mixing for two hours and ultrasonication for 15 min was not enough to disperse the nanoparticles evenly. On the other hand, the shape of the nanoparticles was preserved, and the nanoparticles were stable; neither the polymer nor the temperature used in the preparation process influenced the shape of the AgNPs. The polymer is inert and does not affect the nanoparticles in any way; it acts as a binder.

The non-homogeneous distribution of nanoparticles in the prepared composite solutions was also reflected in the prepared nanofibers, Figure 8. Electrospinning of the PVA–AgNPs composite solution caused the formation of chains of AgNPs in the PVA matrix fibers, Figure 8. It is also clear that nanoparticles are placed inside the fibers (surrounded by the polymer) and are not uniformly distributed, not only in micro volumes (within fibers) but also in macro volumes of non-woven textiles (not present in all fibers). Similar results (for the distribution of nanoparticles) were also observed in the fibers of sample I (data not shown).

Better results regarding the distribution and the number of nanoparticles were achieved by the dip-coating method. Figure 9 shows the fibers and details of the distribution of AgNPs. The nanoparticles are uniformly distributed over the surface of the fibers. It can be expected that with a uniform distribution of AgNPs and mainly because the nanoparticles are on the surface of the fibers, such a textile will have better toxic properties. In addition, the color of the dip-coated fabric was brighter, Figure 6d, than the non-woven fabric prepared by electrospinning.

### 3.2. Antibiofilm Properties

The main mechanisms of toxicity in AgNPs have been proposed by many authors [31,32] and can be described by three mechanisms: oxidative stress, DNA damage, and reproductive disorders. Recent studies have shown that the release of AgNPs into the environment is increasing, yet there are large gaps in understanding of how these particles are transported through ecosystems and migrate into the food chain and their effect on human health [32]. The commercialization of products containing silver nanoparticles is increasing enormously. Many studies and experiments testing the toxicity of bacteria and viruses have been published, and only a few have tested algae and biofilm-forming organisms, higher plants, or animals.

The antibiofilm activity of AgNPs colloids and prepared PVA–AgNPs nanocomposites were tested on a one-cell green algae *P. kessleri*. The slightly modified disc diffusion test was used, where samples were placed on algal inoculated agar. From both PVA–AgNPs polymer nanocomposites’ (B and I) thin layers and non-woven fabrics (prepared by electrospinning and dip-coating methods), samples were taken. As a control, pure PVA fibers were used. After 14 days, the samples were evaluated, and the presence and the size of the zone of inhibition (ZOI) were measured.

It was found that the pure PVA fibers prepared by electrospinning, Figure 10, did not show any antibiofilm effect. It is clear, that the green algae have overgrown the sample completely. On the other hand, the thin layers of sample I, Figure 10, show clear ZOI (9 mm); a similar result was observed in sample B (data not shown). The size of the inhibition zone of fibers that contain AgNPs (sample B) was from 7 to 9 mm in diameter for samples after electrospinning and dip-coating, respectively. A similar result was observed in sample I (data not shown). The colloidal solution of samples B and I showed a distinctive inhibition zone with a size of approximately 11 mm in both cases. The inhibition zones became larger in this order:

Pure PVA fibers < B, I films < electrospinned B, I fibers < dip-coated B, I fibers < AgNPs colloids.

The reason for such behavior is the placement of nanoparticles and the possibility of their direct interaction with cells. AgNPs colloids showed the most distinct ZOI due to the possibility of direct contact of nanoparticles with algal cells. In the case of the dip-coated fibers, the AgNPs were placed on the surface of the fibers, allowing direct contact with the cells, but since they are polymer-bound and the amount of AgNPs is smaller compared to colloids, the ZOI was also smaller. On the other hand, in the case of the electro-spun fibers, the AgNPs were completely surrounded by the polymer matrix, therefore the contact of the nanoparticles with the cells was slowed down by the rate of their release from the polymer matrix. In the case, for a thin film, the nanoparticles are located inside in polymer and form clusters; in addition, the reaction surface of the thin films is much smaller than in the case of the fibers; therefore, it is logical that the samples of thin films show the smallest ZOI. Pure PVA is non-toxic, so it shows no ZOI.

This study has shown that all of the prepared PVA–AgNPs nanocomposites are able to induce long-term algal toxicity. That means that silver nanoparticles transferred their toxic properties into non-toxic PVA and that polymer nanocomposites doped by AgNPs can release AgNPs or silver ions into the environment. The acquirement of antibacterial, toxic, or antibiofilm effects offers a more comprehensive range of uses for polymers. Such polymer composites with antibacterial properties can be used in medical applications; for instance, in work [32], a cellulose nanofiber with AgNPs with antimicrobial activity for developing scaffolds for wound repair was successfully made and analyzed. Our antibacterial results are also in good agreement with other works [9,13,33,34].

The extension of toxic properties to non-toxic polymers is useful in some applications in medicine, e.g., in the protection of water management facilities against the formation of biofilms or in the production of filters for water purification. On the other hand, the goal of future nanotoxicological research must be to create better models for defining toxicity (today most of the tests are performed in vitro on cell cultures with lower life forms) and to develop efficient recycling technologies for such products.

## 4. Conclusions

AgNPs of different shapes were successfully prepared by the chemical reduction method. Different shapes of nanoparticles result in different colors of colloidal solutions. The blue solution contains triangular nano-prisms with an average size of 75 nm. The triangular nanoparticles were stable during the experiment. The second colloid changed color from yellow (day 1) to violet on day 25. The yellow solution contained spherical nanoparticles (average size 11 nm) that change into a mixture of different shapes (irregular, triangular, rod-shaped, spherical, decahedral, etc.) with time. Such a mixture of nanoparticle shapes led to the dichroic effect of the colloid.

We successfully incorporated pre-prepared AgNPs into the PVA polymer matrix using an ex situ method. The prepared composites’ thin layers (by spin-coating) and fibers (by electrospinning and dipping) took over the optical and toxic properties of the silver nanoparticles.

Thin films and non-woven fabrics of polymers doped with AgNPs could be used to protect surfaces in water and humid environments, in the water industry, in medical applications (vascular stents, cartilages, catheters, contact lenses, etc.), and in food packaging.

## Figures and Tables

**Figure 1 polymers-15-00379-f001:**
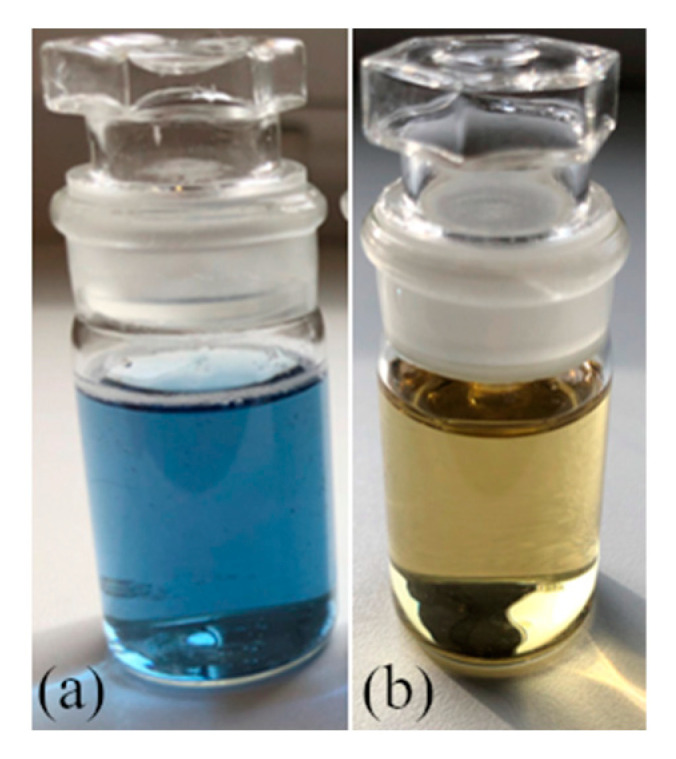
Colloidal solution: sample B (**a**) and I (**b**).

**Figure 2 polymers-15-00379-f002:**
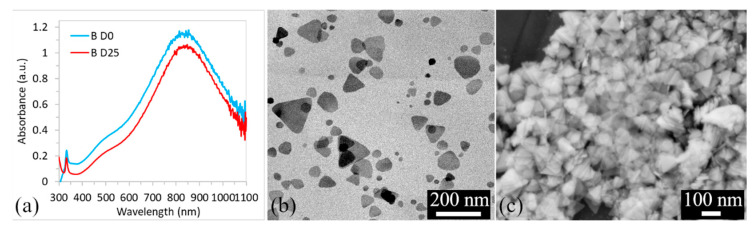
UV–vis spectra of the AgNPs colloids sample B for the day of synthesis (D0) and the 25th day (D25) (**a**); TEM microphotograph (**b**) and SEM (**c**) of AgNPs on D0.

**Figure 3 polymers-15-00379-f003:**
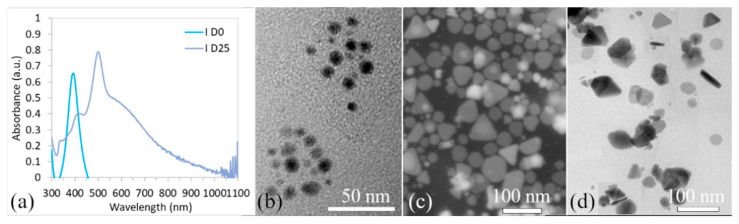
UV–vis spectra of the AgNPs colloid (I sample) for the day of synthesis (D0) and day 25 (D25) (**a**); AgNPs TEM micrographs at D0 (**b**) and TEM and SEM micrographs at D25 (**c,d**).

**Figure 4 polymers-15-00379-f004:**
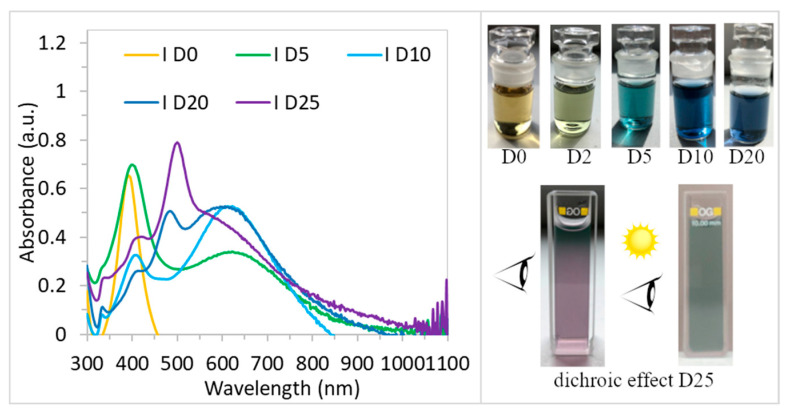
UV–vis spectra of AgNPs colloids (sample I) and colloid color changes with time.

**Figure 5 polymers-15-00379-f005:**
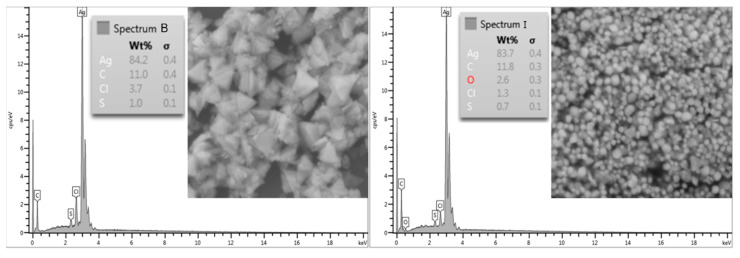
The energy-dispersive X -ray spectroscopy of both samples B and I.

**Figure 6 polymers-15-00379-f006:**
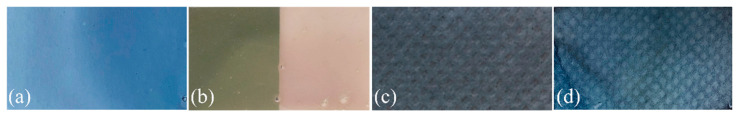
The PVA–AgNPs thin film: sample B (**a**) and the sample I (**b**); nanofibers of non-woven textile prepared by electrospinning (**c**) and dip-coating (**d**) of sample B.

**Figure 7 polymers-15-00379-f007:**
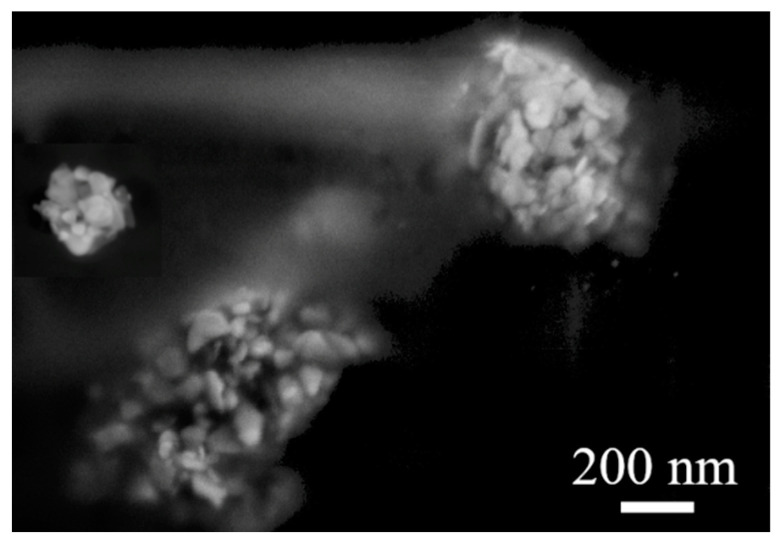
SEM microphotograph of AgNPs clusters in the PVA–AgNPs thin film, sample B.

**Figure 8 polymers-15-00379-f008:**
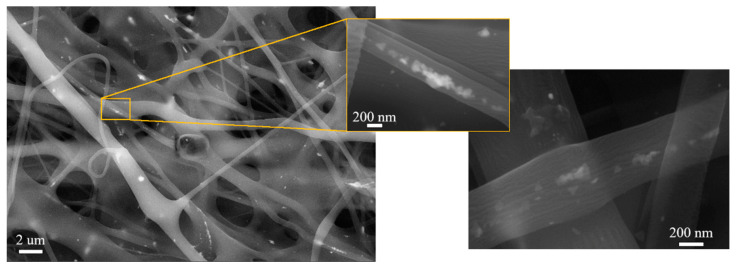
SEM microphotograph of AgNPs clusters in PVA–AgNPs nanofibers of non-woven textiles, sample B, and the detail of the nanoparticles in the fibers.

**Figure 9 polymers-15-00379-f009:**
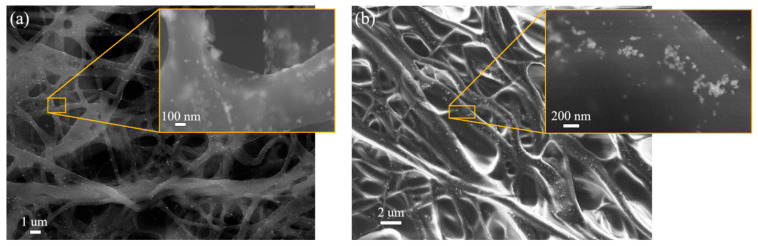
SEM microphotograph of PVA–AgNPs composite. Dip-coating technique, sample B (**a**) and the sample I (**b**).

**Figure 10 polymers-15-00379-f010:**
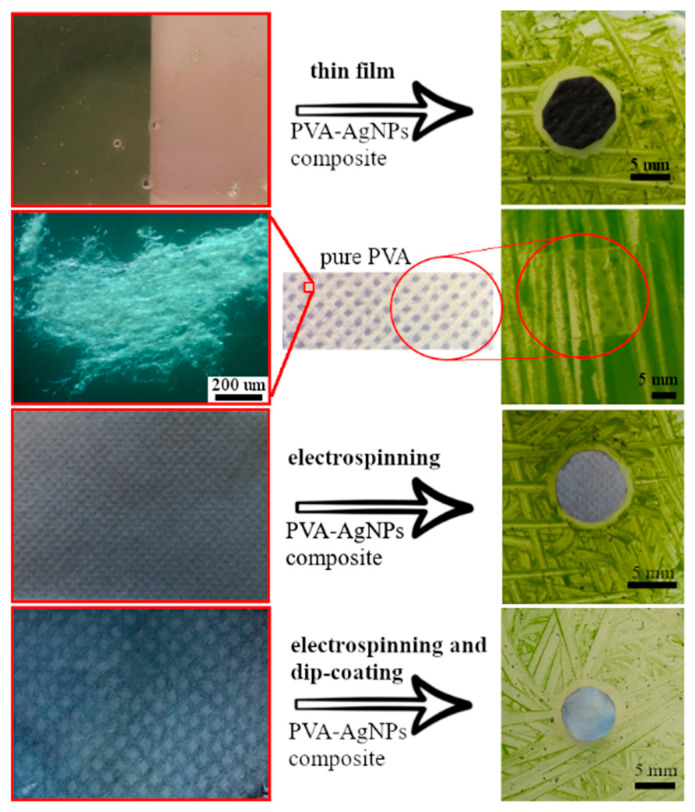
The anti-biofilm effect of the PVA–AgNPs composites’ thin layer (sample I); of pure PVA fibers and PVA–AgNPs composite fibers (sample B) prepared by electrospinning and dip-coating (shown from top to bottom).

**Table 1 polymers-15-00379-t001:** Concentrations of the reagents.

Reagents	Amount (mL)		Concentration
	B	I	
Silver nitrate	4300	4300	0.11 mM
Sodium citrate	368	368	30 mM
Polyvinylpyrrolidone	368	368	2% *w*/*w*
Hydrogen peroxide	12	-	30% *w*/*w*
Sodium borohydride	20	20	0.1 M

## Data Availability

Not applicable.

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
