# Peer review of "Preparation, Structure, and Properties of PVA–AgNPs Nanocomposites"

_polymers, 2023, doi:10.3390/polym15020379_

Round 1

Reviewer 1 Report

The work provides important information for promoting biopolymer matrix composite preparation based on silver nanoparticle and their toxic properties. The data in this manuscript is well elaborated. I recommend the publication of this manuscript in Polymers Journal. 

Author Response

Dear Reviewer,

thank you for your rating. We are very pleased with such a positive evaluation.

With best regards authors.

Reviewer 2 Report

This article, a Velgosova et al., titled Incorporation of AgNPs into the polymer matrix" is interesting and will definitely advance the field. However, revisions are required to improve the article structure as follows:

1)     The title should be modified to a more appealing one and include the work's actual application/aim.

2)     In line 22, please correct “originally”.

3)     What is the actual significance of adding silver NPs to the polymeric matrix? This should be explained and highlighted in the abstract, and results and discussion.

4)     In line 93, the authors mentioned “to observe the synergic effect of polymer and AgNPs”. Synergistic effect on what? Please refer to point 3.

5)     Based on points 3 and 4, the authors should conduct at least two antibacterial assays (MIC, or others) to compare the antibacterial effects of silver NPs to PVA-AgNPs.

6)     Also, the authors should test the antibiofilm of the prepared NPs/nanofibers.

7)     Table 1 is vague. Please write a short description of how these different solutions were prepared. What is the concentration of each compound? The final concentration is for which solution?

8)     The authors should describe the detailed reduction method with suitable citations.

9)     More chemical analysis should be conducted to confirm the successful synthesis of Ag NPs and PVA-AgNPs, such as XRD and FTIR.

10)  The authors should correlate and compare their findings to previous reports concerning loading silver NPs into polymeric matrices.

Author Response

Dear Reviewer,

thank you for your rating. Answers to comments are in the attached file.

Best regards, authors.

Reviewer 3 Report

In this manuscript, the author prepared two different colloids of AgNPs firstly, then PVA-AgNPs composites were prepared by ex-situ method. In addition, the obtained PVA-AgNPs composites not only showed a dichroic effect, but also has good antibacterial effect. There some issued need to be addressed, therefore a major revision of this review is recommended.

(1) The title should be revived, such as Preparation, structure, and properties of Ag/PVA nanocomposites.

(2) The PVA is not PLA, PVA was obtained from the hydrolysis of polyvinyl acetate, which is a biopolymer, thus the “biopolymer matrix composite” in line 11 should be revised.

(3) The abbreviation such as PVA-AgNPs should be defined when it appeared for the first time.

(4) What is the novelty for this work? To highlight innovation, the author can emphasis the effect of AgNPs shapes on the properties in the abstract.

(5) Typical synthesis procedure and corresponding references should provided in the section of synthesis of silver nanoparticles

(6) Does the AgNPs colloids stable at high temperature? As the composites was obtained at 85ºC.

(7) The optical properties and morphology of the obtained AgNPs colloids should compared with the result in the literature.

(8) What is the EM type for Figure 6,7,8?

(9) The antibacterial properties of the obtained composites should compared with the result in the literature.

Author Response

Dear Reviewer,

thank you for your comments all answers are in the attached file.

Best regards, authors.

Round 2

Reviewer 2 Report

The authors responded very well to the reviewers' comments. I recommend the publication of the paper in its current form.

Reviewer 3 Report

All of the issues mentioned were resolved in detail